# MicroRNA profiling in umbilical cord plasma: links to maternal metabolism and neonatal metabolic and inflammatory traits

Jasmin Zaunschirm-Strutz[1,2] (ID), Anna Rieder[2,3], Carolina Tocantins[1,4,5,6] (ID), Mariana S. Diniz[1,4,5,6] (ID), Elisa Weiss[1] (ID) and Ursula Hiden[1] (ID)

[1]*Department of Obstetrics and Gynecology, Research Unit Early Life Determinants (ELiD), Medical University of Graz, Graz, Austria*

[2]*Institute of Biomedical Sciences, Carinthian University of Applied Science, Klagenfurt, Austria*

[3]*Division of Physiology and Pathophysiology, Cardio-Metabolic Research, Medical University of Graz, Graz, Austria*

[4]*CNC-UC-Center for Neuroscience and Cell Biology, University of Coimbra, Coimbra, Portugal*

[5]*Center for Innovative Biomedicine and Biotechnology (CIBB), University of Coimbra, Coimbra, Portugal*

[6]*Doctoral Programme in Experimental Biology and Biomedicine (PDBEB), Institute for Interdisciplinary Research (IIIUC), University of Coimbra, Coimbra, Portugal*

Handling Editors: Laura Bennet & Janna Morrison

The peer review history is available in the Supporting Information section of this article (https://doi.org/10.1113/JP287672#support-information-section).

Circulating umbilical cord plasma miRNAs profiles are shaped by maternal and neonatal metabolic traits

**Abstract figure legend** The present study explored the relationship between circulating microRNAs (miRNAs) in umbilical cord plasma (UCP), analysed via next-generation sequencing, and maternal metabolic traits, in addition to neonatal anthropometric, metabolic and inflammatory traits. Correlation analysis identified maternal body mass index (BMI) and gestational weight gain as the strongest influencing factors among maternal characteristics. For neonatal characteristics, placental weight, neonatal glucose metabolism and UCP leptin levels showed the most significant effects. The font size illustrates the number of miRNAs correlating with these key influencing factors. Among the miRNAs, the pregnancy-specific miRNA clusters located on chromosomes 19 (C19MC) and 14 (C14MC) were particularly affected.

**Jasmin Zaunschirm-Strutz** is a university lecturer and PhD candidate with a strong research interest in maternal metabolic disorders during pregnancy and their impact on fetal programming. Her research focuses on the role of circulating microRNAs in maternal and neonatal health, with a particular emphasis on those found in umbilical cord blood and in fetoplacental endothelial cells. As a placental researcher, she aims to elucidate the functional impact of dysregulated microRNAs on fetoplacental endothelial cells, using advanced *in vitro* angiogenesis assays. Through this research, she strives to advance the understanding of the molecular mechanisms linking maternal metabolism to neonatal outcomes.

**Abstract** MicroRNAs (miRNAs) are regulators of mRNA translation and play crucial roles in various physiological and pathological processes. In this study, we profiled miRNAs in umbilical cord plasma (UCP) to explore the association of neonatal circulating miRNAs with maternal metabolic parameters and neonatal anthropometric, metabolic and inflammatory characteristics in healthy pregnancies. Data and UCP samples were collected from 16 pregnancies, equally divided between normal-weight and overweight mothers and between male and female newborns. Using next-generation sequencing, we identified and quantified miRNAs in UCP, alongside the analysis of metabolic and inflammatory parameters. Our results revealed that the majority of UCP miRNAs are sensitive to maternal and neonatal characteristics, particularly maternal body mass index, gestational weight gain, placental weight, UCP leptin, UCP C-reactive protein and UCP insulin levels. Notably, we identified a strong association between the placenta-derived chromosome 19 microRNA cluster (C19MC) and placental weight, gestational weight gain, UCP insulin and neonatal weight. Likewise, the pregnancy-specific chromosome 14 microRNA cluster (C14MC) was associated with maternal body mass index and UCP leptin. Our study highlights the sensitivity of UCP miRNAs to maternal metabolic conditions, demonstrates their association with neonatal metabolic and inflammatory traits, and underscores the potential role of circulating cord blood miRNAs in fetal metabolism and development.

(Received 10 September 2024; accepted after revision 5 February 2025; first published online 27 February 2025)

**Corresponding author** Ursula Hiden: Department of Obstetrics and Gynecology, Medical University of Graz Auenbruggerplatz 14, 8036 Graz, Austria. Email: ursula.hiden@medunigraz.at

## Key points

- MicroRNAs (miRNAs) are regulatory RNA molecules that modulate protein expression. They are present in all body fluids and umbilical cord plasma and are affected by metabolic changes.
- Pregnancy is a state of metabolic change in the mother, and maternal metabolism affects fetal development.
- We found that the composition of umbilical cord blood miRNAs is associated with maternal and neonatal metabolism.
- Pregnancy-specific groups of miRNAs showed particular patterns, with miRNAs encoded by a region of chromosome 14 associated with maternal body mass index and with miRNAs encoded by a specific region of chromosome 19 associated with umbilical cord plasma insulin.
- MicroRNAs represent a separate dimension through which maternal metabolism can influence fetal development.

## Introduction

MicroRNAs (miRNAs) are small RNA molecules ∼22 nucleotides in length that modulate the translation of homologous target mRNAs (Gebert & MacRae, 2019). Thus, miRNAs play a role in various physiological and pathological processes, including cell death, stress response, metabolism, cell differentiation and proliferation (Bartel, 2004). In humans, the miRNA database miRBase (www.mirbase.org) (Kozomara et al., 2019) currently lists 1917 precursor miRNAs, each of which usually produces two mature miRNAs. Given that most mammalian genes are targets for miRNA binding (Friedman et al., 2009), miRNAs represent a complex system for the epigenetic regulation of protein expression.

MicroRNAs are found not only in the cytoplasm but also in body fluids (Weber et al., 2010). In plasma, miRNAs are typically associated with lipid and protein carriers that protect them from degradation (Vickers et al., 2011). Altered levels of circulating miRNAs have been implicated in pathological conditions, such as cancer (Pozniak et al., 2022), and in metabolic disorders, such as high body mass index (BMI) and insulin resistance (Ji & Guo, 2019, Jones et al., 2017).

Maternal metabolism influences fetal metabolism and development profoundly by regulating the delivery of key nutrients, such as glucose and lipids. For instance, maternal overweight and obesity are associated with higher birth weight, increased neonatal body fat (Friis et al., 2013; Wan et al., 2023) and fetal insulin resistance

(Catalano et al., 2009). Conversely, pregnancy presents a challenge to the maternal metabolism, initiating a series of metabolic adaptations driven by pregnancy hormones, growth factors and other regulatory molecules, many of which are released by the placenta (Stern et al., 2021). These bioactive factors play a crucial role in adapting maternal metabolism to meet the demands of pregnancy and ensuring an adequate supply of nutrients to the developing fetus (Stern et al., 2021). As such, maternal BMI is likely to be one of the most significant factors influencing fetal development, fundamentally shaping fetomaternal communication and potentially impacting umbilical cord plasma (UCP) miRNA patterns.

Interestingly, the release of various placenta-derived hormones and growth factors differs depending on fetal sex, leading to distinct levels of certain hormones and growth factors, such as progesterone, placental growth hormone and human chorionic gonadotropin, in the maternal circulation (Stern et al., 2021). Moreover, some bioactive factors in the fetal circulation, such as leptin (Tome et al., 1997), also vary between the sexes. Most notably, neonatal fat mass differs significantly between male and female neonates (Rodriguez et al., 2004; Shields et al., 2006), positioning fetal sex as a potential factor influencing fetomaternal communication.

In this study, we aimed to investigate the association between maternal metabolic characteristics and UCP miRNAs in uncomplicated, healthy pregnancies. Furthermore, we investigated whether UCP miRNAs interact with neonatal anthropometric, metabolic and inflammatory traits. To this end, we established a cohort of UCP samples from pregnancies of women who were normal weight or overweight. We analysed UCP miRNAs using next-generation sequencing in relationship to maternal BMI, glycaemia and gestational weight gain and determined their associations with neonatal anthropometrics, sex, and metabolic and inflammatory parameters.

## Methods

### Ethical approval

Ethical approval was obtained from the Ethics Committee of the Medical University of Graz (29-319ex16/17), and the study was performed in accordance with the latest revision of the *Declaration of Helsinki* (version 2024), except for registration in a database. All women participating in the study provided written informed consent.

### Collection of UCP

Venous cord blood was collected from 30 healthy full-term pregnancies. Umbilical cord blood of 16 women was

selected in order to achieve a wide range of maternal pre-pregnancy BMI, i.e. eight women with a normal pre-pregnancy BMI of $<25$ kg/m$^2$ and eight women with a high pre-pregnancy BMI of $>25$ kg/m$^2$. Moreover, we selected women in order to include a similar number of pregnancies with male and female fetuses in both (normal and high pre-pregnancy BMI) groups. Cord blood was obtained immediately after delivery using Vacuette EDTA blood collection tubes (Greiner Bio-One, Kremsmünster, Austria). Plasma was obtained after centrifugation at 2000 *g* and 4°C for 10 min and stored at $-80$°C for analysis.

### MicroRNA extraction

MicroRNA extraction from UCP was performed using the miRNeasy Serum/Plasma Advanced Kit (Qiagen, Hilden, Germany). Optional on-column DNase digestion was performed using RNase-Free DNase Set (Qiagen) to improve sequencing quality. For the extraction, 200 µl of UCP was used, and the final elution volume was 20 µl of RNase-free water. The quality of the collected miRNA was assessed using the Small RNA Kit on the Bioanalyzer (Agilent, Santa Clara, CA, USA). It covers the size range of 6–150 nucleotides and allows quantification in the range of 50–2000 pg/µl, using only 1 µl of RNA. The extracted RNA was quantified using the Qubit$^{TM}$ microRNA assay kit (Thermo Fisher Scientific, Waltham, MA, USA) and the Quantus$^{TM}$ fluorometer (Promega, Fitchburg, WI, USA), and concentrations ranged from 0.55 to 1.88 ng/µl.

### Library preparation

For library preparation, the QIAseq® miRNA UDI Library Kit (Qiagen) was used. The kit supports RNA input amounts of 1–500 ng total RNA in 5 µl, through precise adapter and primer dilution specifications. First, a pre-adenylated DNA adapter was bound to the 3′ end of miRNAs, followed by a second reaction in which an RNA adapter was bound to the 5' end. Universal complementary DNA (cDNA) synthesis was then performed using primers to add a unique molecular identifier to each molecule. Magnetic bead purification of the cDNA was followed by library amplification, in which the i5 and i7 indices were added to the cDNA, giving each reaction a unique dual index. After a second magnetic bead purification step, quality control was assessed using the Agilent High Sensitivity DNA Kit on the Bioanalyzer (Agilent). Library quantification was performed on a Quantus fluorometer with the QuantiFluor® ONE dsDNA System (Promega).

### Sequencing

Libraries were diluted to a final concentration of 4 nM and pooled. Five microlitres of the equimolar pooled

libraries was denatured with 5 μl of 1 м NaOH. After adding 5 μl of 200 mM Tris-Cl pH 7 and the diluted HT1 hybridization buffer (provided by Illumina), the final library concentration was 1.2 pM. Sequencing was performed using the MiniSeq™ High Output Reagent Kit (75 cycles; Illumina) on the MiniSeq System (Illumina). The sequences of indexes and sample assignments were imported into the Local Run Manager software (Illumina) using a CSV template. Sequencing conditions were set to FASTQ only, single read 72 bp and dual indexes (10 bp). Four UCP samples and a human brain total RNA control (Thermo Fisher Scientific, Waltham, MA, USA) were analysed per sequencing run. Given that a total read count of 25 million reads is possible per run, this allows a maximum read count of 5 million reads per sample.

Primary analysis of miRNA sequencing was performed on GeneGlobe software (Qiagen) using the RNA-seq Analysis & Biomarker Discovery Pipeline. The FASTQ files were uploaded on the RNA-seq Analysis Portal 4.0 (Qiagen) and aligned against a reference [miRBase_v22, homo sapiens (GRCh38.103)]. In addition, a trimmed mean normalization of M values was performed by the software to eliminate differences in sequencing depth. All samples with a count per million (CPM) > 10 were then processed further for statistical analysis.

### Analysis of UCP parameters

Umbilical cord plasma samples were thawed on ice and analysed using Cobas e411 and Cobas c111 analysers (Roche Diagnostics, Basel, Switzerland). Prior to measurement, the different analytes were calibrated, and the accuracy was verified using two quality checks. The e411 analyser measured interleukin 6 (IL6), human growth hormone (hGH), C-peptide and insulin, whereas the c111 instrument measured cholesterol, triglycerides, high-density lipoprotein (HDL), low-density lipoprotein (LDL), C-reactive protein (CRP) and glucose. Tumor necrosis factor alpha (TNF$\alpha$) levels were measured using the Human TNF-alpha Quantikine HS ELISA (R&D Systems, Minneapolis, MN, USA) according to the manufacturer's instructions.

### Data analysis

Statistical analysis of maternal and neonatal characteristics was performed using GraphPad Prism v.10.3.0 and IBM SPSS Statistics (v.29.0.0.0) software packages. Data were tested for normal distribution using the Shapiro–Wilk test. Accordingly, parametric and non-parametric tests were used for group comparisons (Student's unpaired *t* test and the Mann–Whitney *U*-test) and correlation analyses (Pearson's and Spearman's correlations). Correlation analysis of miRNAs with maternal and neonatal characteristics used Python 3.13 software. After testing for normal distribution using the Shapiro–Wilk test, Pearson's or Spearman's correlation was applied. For correlation analysis of miRNAs with nominal data (male/female) point-biserial correlation was performed. Cluster analysis and heatmap generation were performed using the ClustVis 2.0 online tool (https://biit.cs.ut.ee/clustvis/; Metsalu & Vilo, 2015). Pathway analyses of chromosome 14 miRNA cluster (C14MC) miRNAs and of miRNAs differentially expressed between UCP of male and female neonates were performed using miRPath 4.0 (http://www.microrna.gr/miRPathv4; Tastsoglou et al., 2023). Potential targets of chromosome 19 miRNA cluster (C19MC) miRNAs were obtained from the TargetScan v.8.0 online tool (www.targetscan.org).

## Results

### Cohort characteristics

To investigate the relationship between UCP miRNA signatures and maternal and neonatal characteristics, we established a cohort of UCP samples from healthy, uncomplicated pregnancies of 16 women. Eight women had a pre-pregnancy BMI of <25 kg/m$^2$ and eight women had a pre-pregnancy BMI of >25 kg/m$^2$. The cohort consisted equally of pregnancies of normal-weight and overweight women and of pregnancies with male and female newborns. The following data were collected or analysed from these mother–child dyads: maternal anthropometry, age and oral glucose tolerance test (oGTT) values at mid-pregnancy, neonatal anthropometry and the following UCP parameters: glucose, insulin and C-peptide, inflammatory markers (CRP, IL6 and TNF$\alpha$), hGH, leptin and lipids (HDL, LDL, cholesterol and triglycerides). Based on UCP glucose and UCP insulin, we calculated the neonatal homeostatic model assessment of insulin resistance (HOMA-IR; Hancox & Landhuis, 2011). The characteristics of the subjects are summarized in Table 1.

### Umbilical cord plasma miRNA profiling

MicroRNA sequencing identified a total of 1562 individual miRNAs in UCP. Given that the majority of these miRNAs were present at low levels, we focused our data analysis on miRNAs with a mean signal of >10 counts per million (CPM) across all 16 samples. This threshold was met by 387 miRNAs. Among the detected miRNAs were members of the C19MC and the C14MC. The C19MC, expressed predominantly in the placenta, comprises 46 precursor miRNAs (pre-miRNAs; Bortolin-Cavaille et al., 2009; Paquette

**Table 1. Subject characteristics**

| Characteristic | Value |
|---|---|
| Sample size | 16 |
| Mode of delivery (vaginal/caesarean section) | 5/11 |
| Pregnancy length (weeks) | $39.3 \pm 0.9$ |
| Maternal parameters | |
| Maternal age (years) | $29.4 \pm 4.8$ [21–39] |
| Pre-pregnancy BMI (kg/m$^2$) | $24.9 \pm 4.1$ [19.2–33.3] |
| BMI at birth (kg/m$^2$) | $29.0 \pm 4.4$ [23.3–38.0] |
| Gestational weight gain (kg) | $12.1 \pm 6.9$ [1.0–25.0] |
| oGTT 0 h (mg/dl) | $79.3 \pm 8.1$ [62.0–91.0] |
| oGTT 1 h (mg/dl) | $122.0 \pm 28.7$ [83–176] |
| oGTT 2 h (mg/dl) | $107.2 \pm 24.9$ [70–142] |
| Neonatal parameters | |
| Placental weight (g) | $632 \pm 143$ [430–920] |
| Neonatal weight (g) | $3390 \pm 388$ [2870–4390] |
| Fetoplacental weight ratio | $5.51 \pm 0.77$ [4.31–7.47] |
| Neonatal length (cm) | $50.5 \pm 2.8$ [47–39] |
| **Umbilical cord plasma parameters** | |
| Glucose (mg/dl) | $69.5 \pm 15.2$ [36.7–105.7] |
| C-peptide (pmol/l) | $240 \pm 68$ [106–362] |
| Insulin (µU/ml) | $9.9 \pm 5.8$ [2.4–23.4] |
| HOMA-IR | $1.80 \pm 1.42$ [0.2–6.1] |
| CRP (ng/ml) | $82.5 \pm 21.1$ [40–130] |
| IL6 (pg/ml) | $7.91 \pm 3.91$ [2-39–17.31] |
| TNF$\alpha$ (pg/ml) | $0.912 \pm 0.156$ [0.633–1.196] |
| hGH (µg/ml) | $12.46 \pm 7.95$ [0.08–30.92] |
| Leptin (ng/ml) | $2.28 \pm 1.49$ [0.51–4.97] |
| HDL (mmol/l) | $0.71 \pm 0.16$ [0.44–1.08] |
| LDL (mmol/l) | $0.52 \pm 0.24$ [0.17–1.10] |
| Cholesterol (mg/dl) | $57.0 \pm 15.5$ [27.3–86.9] |
| Triglycerides (mg/dl) | $33.4 \pm 12.5$ [15.0–59.2] |

Data are presented as the mean $\pm$ SD, with the range in square brackets. Abbreviations: BMI, body mass index; CRP, c-reactive protein; HDL, high-density lipoprotein; hGH, human growth hormone; HOMA-IR, homeostatic model assessment for insulin resistance; IL6, interleukin 6; LDL, low-density lipoprotein; oGTT, oral glucose tolerance test; TNF$\alpha$, tumor necrosis factor alpha.

et al., 2018). Of these, 39 mature miRNAs derived from 30 distinct C19MC-encoded pre-miRNAs were identified in UCP with signals of >10 CPM. Likewise, the C14MC, which is also pregnancy specific, encodes 56 pre-miRNAs (Paquette et al., 2018). In UCP, 55 mature miRNAs originating from 36 distinct C14MC-encoded pre-miRNAs were detected.

To investigate the influence of maternal and neonatal factors on UCP miRNAs, we performed a correlation analysis. Of the 387 miRNAs with signals of >10 CPM, 255 miRNAs (65.9%) showed significant correlations with maternal and/or neonatal characteristics. Notably, maternal parameters, such as pre-pregnancy BMI, BMI at birth and gestational weight gain (GWG), demonstrated strong associations with miRNA signatures. Likewise, neonatal parameters, including placental weight, fetoplacental weight ratio, UCP leptin, UCP CRP and

UCP insulin, exhibited strong correlations with specific miRNA (Fig. 1). A complete list of miRNAs correlated with individual subject characteristics is provided in Supplementary Table 1.

Among the miRNAs associated with neonatal anthropometrics, placental weight, neonatal glucose metabolism and GWG, a significant proportion belonged to the C19MC. In contrast, C14MC miRNAs were correlated predominantly with maternal BMI (both pre-pregnancy and at birth) and with UCP leptin and UCP CRP levels.

## Cluster analysis

To identify groups of miRNAs that show similar associations with maternal and neonatal characteristics, on the one hand, and in order to identify groups of

maternal or neonatal characteristics that show similar associations with UCP miRNAs, on the other hand, we performed cluster analysis. Therefore, we used the correlation coefficients of maternal and neonatal characteristics with each individual miRNA. Only the 255 miRNAs significantly correlated with at least one maternal or neonatal parameter were included in the cluster analysis. MicroRNAs without any associations were excluded. The analysis generated a heatmap that visualizes the correlations, with red and blue colours indicating positive and negative correlation coefficients, respectively. The dendrograms above and to the left of the heatmap illustrate the clustering of maternal and neonatal characteristics and miRNA signatures, respectively. The short branches of the top dendrogram show that pre-pregnancy BMI, BMI at birth, oGTT at 0 h and UCP leptin clustered closely together, as did placental weight and GWG, indicating similar associations with UCP miRNAs. Conversely, specific groups of miRNAs clustered together, indicating similar

associations with maternal and neonatal parameters. The clustering generated two main branches of miRNAs, which were separated into Fig. 2*A* and Fig. 2*B* for clarity. The first branch (Fig. 2*A*) included 120 miRNAs and highlighted a prominent cluster of miRNAs positively associated with neonatal HOMA-IR, UCP insulin and fetoplacental weight ratio, but negatively associated with placental weight. This cluster was composed almost entirely of miRNAs from the C19MC, which were marked in green. Another major cluster within this first branch showed positive associations with pre-pregnancy BMI, BMI at birth and UCP leptin and included various C14MC miRNAs, marked in yellow, which, however, did not cluster together as C19MC miRNAs. The second major branch of miRNAs (Fig. 2*B*) included 135 miRNAs and did not reveal distinct large clusters of associations. Only a smaller group of miRNAs exhibited consistent associations with maternal BMI, oGTT at 0 h and UCP leptin.

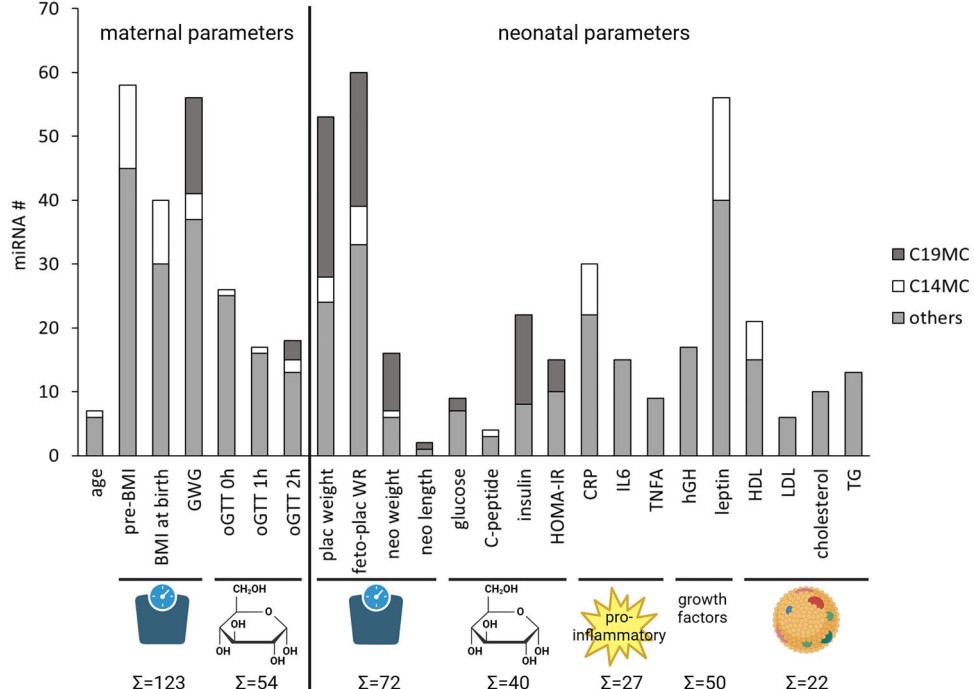

**Figure 1. Number of umbilical cord plasma microRNAs correlating with maternal and neonatal characteristics**
Only the 387 miRNAs with a mean signal of >10 CPM were used for correlation analysis. Maternal BMI, GWG, oGTT at 0 h, placental weight, fetoplacental weight ratio, UCP insulin, UCP CRP and UCP leptin were particularly associated with a high number of UCP miRNAs. The proportions of miRNAs of the C19MC and C14MC are highlighted in dark grey and white, respectively. To simplify interpretation, related characteristics were grouped into broader categories: weight-related characteristics, glucose metabolism, pro-inflammatory markers, growth factors and lipids. The number of miRNAs associated with each condensed group is indicated below. Abbreviations: C14MC, chromosome 14 miRNA cluster; C19MC, chromosome 19 miRNA cluster; CPM, counts per million; feto-plac WR, fetoplacental weight ratio; GWG, gestational weight gain; miRNA, microRNA; neo length, neonatal length; neo weight, neonatal weight; plac weight, placental weight; Pre-BMI, pre-pregnancy BMI; TG, triglycerides; UCP, umbilical cord plasma. [Colour figure can be viewed at wileyonlinelibrary.com]

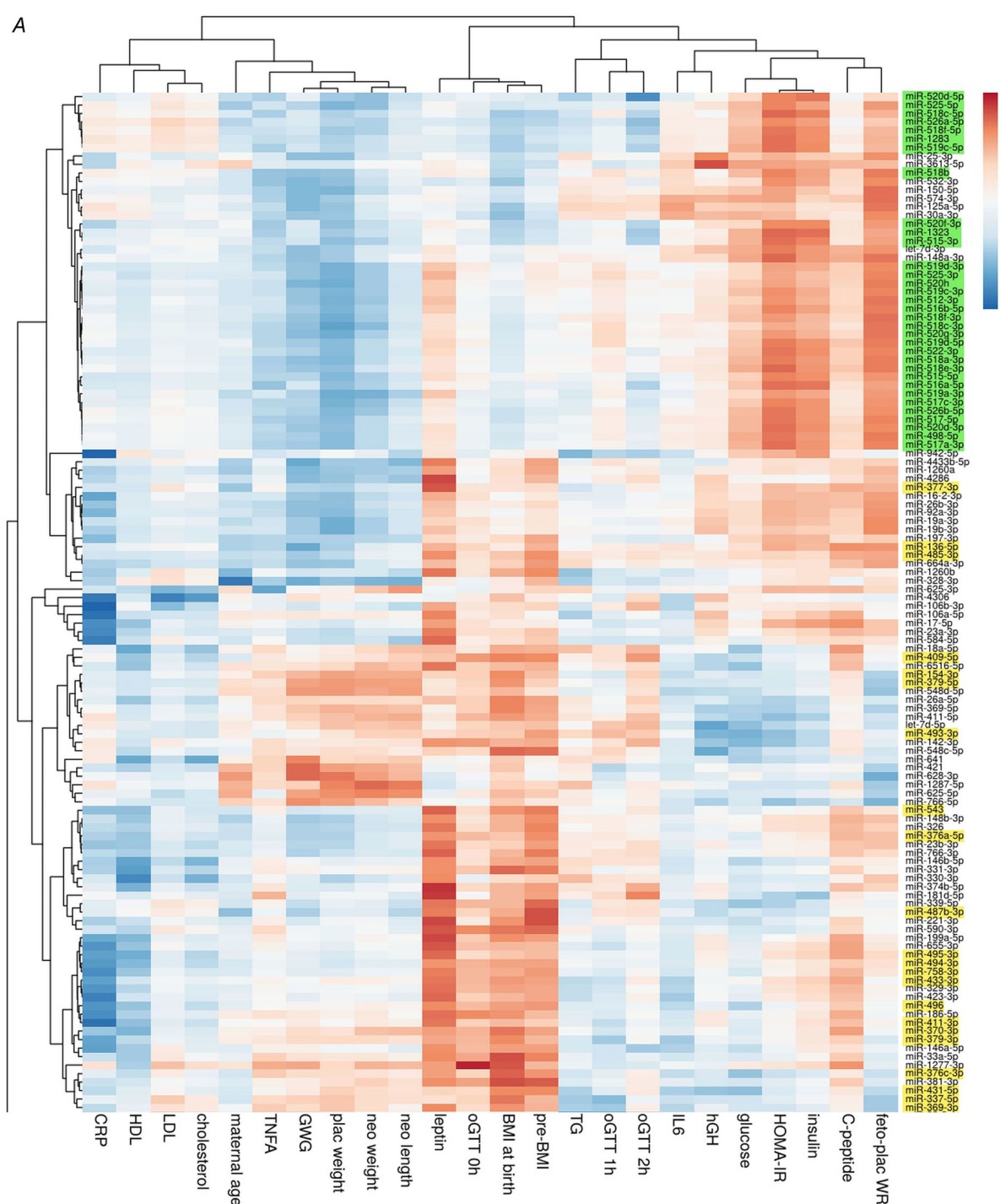

**Figure 2. Heatmap and cluster analysis of correlations between maternal and neonatal characteristics and umbilical cord plasma microRNAs**

Only miRNAs with a mean signal of >10 CPM and correlating with at least one of the maternal or neonatal characteristics were included. Correlation coefficients were used to generate the heatmap and to cluster the maternal and neonatal characteristics (at the top) and the miRNAs (on the left) performed by correlation distance clustering and McQuitty linkage via the ClustVis online tool. MicroRNAs of the C19MC are highlighted in green, miRNAs of the C14MC are highlighted in yellow. *A*, the first main branch of miRNAs encompasses all C19MC miRNAs and most C14MC miRNAs. *B*, the second main branch of miRNAs. Abbreviations: C14MC, chromosome 14 miRNA cluster; C19MC, chromosome 19 miRNA cluster; CPM, counts per million; Pre-BMI: pre-pregnancy BMI; GWG: gestational weight gain; miRNA, microRNA; plac weight: placental weight; feto-plac WR: fetoplacental weight ratio; neo weight: neonatal weight; neo length: neonatal length; TG: triglycerides. [Colour figure can be viewed at wileyonlinelibrary.com]

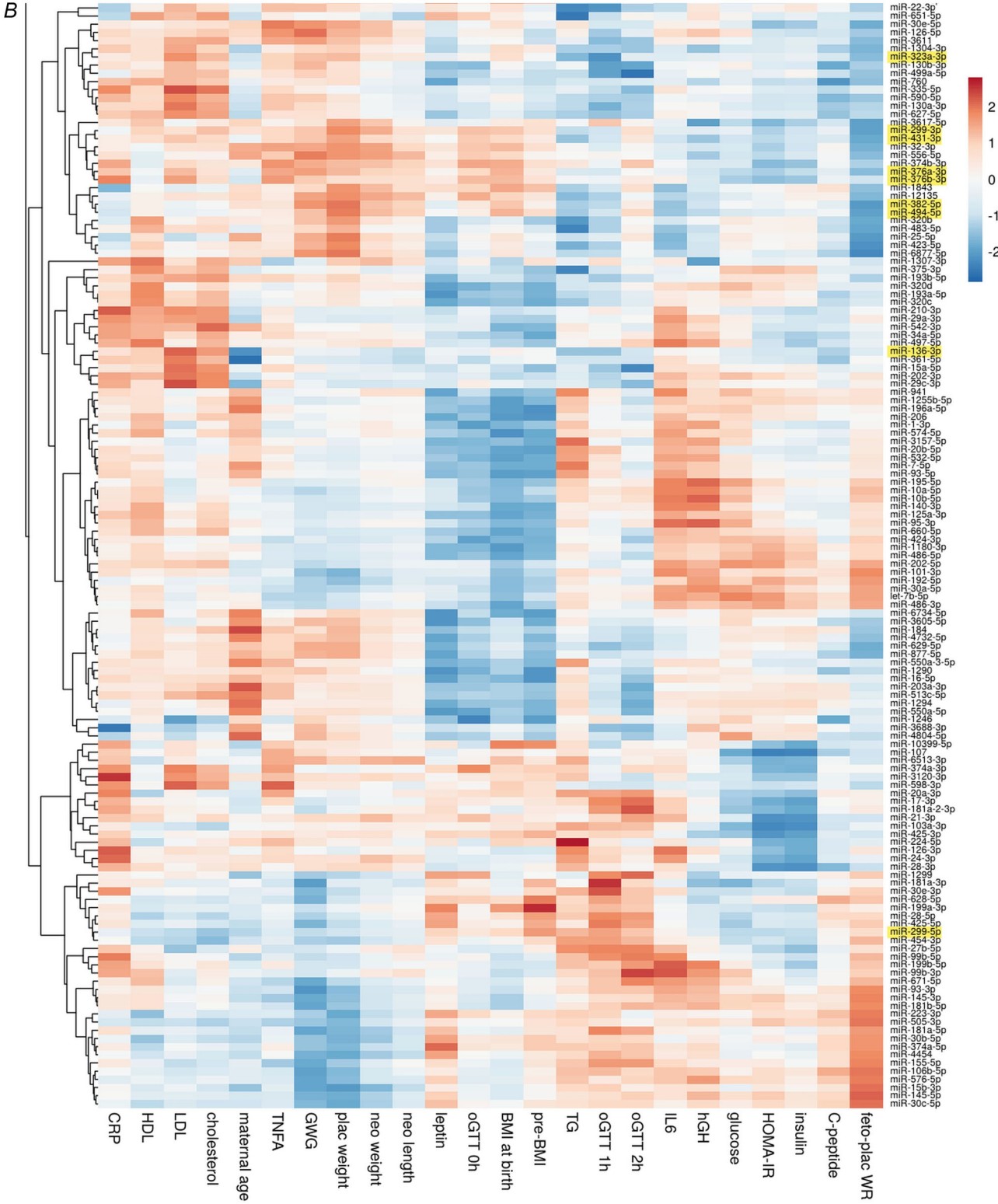

**Figure 2. Continued**

**Table 2. Correlation of chromosome 19 miRNA cluster microRNAs in umbilical cord plasma with placental weight, fetoplacental weight ratio, gestational weight gain, neonatal weight, umbilical cord plasma insulin and umbilical cord plasma homeostatic model assessment of insulin resistance**

| mRNA ID | Mean CPM | Placental weight | Fetoplacental WR | GWG | Neonatal weight | Insulin | HOMA-IR |
|---|---|---|---|---|---|---|---|
| miR-517a-3p * | 4663 | **−0.624** | **0.538** | **−0.542** | **−0.519** | 0.421 | 0.394 |
| miR-516a-5p | 2356 | −0.488 | **0.513** | −0.254 | −0.391 | **0.666** | **0.521** |
| miR-519c-5p * | 2015 | −0.355 | 0.309 | −0.223 | −0.266 | **0.521** | 0.497 |
| miR-517c-3p | 1366 | **−0.591** | **0.598** | −0.405 | **−0.520** | **0.589** | 0.391 |
| miR-515-3p # | 1080 | **−0.518** | **0.556** | −0.302 | −0.433 | **0.681** | **0.585** |
| miR-1323 #* | 1021 | −0.474 | **0.549** | −0.264 | −0.354 | **0.740** | **0.632** |
| miR-519a-3p | 799 | **−0.525** | 0.415 | −0.371 | −0.440 | 0.365 | 0.332 |
| miR-518c-3p | 678 | **−0.685** | **0.706** | **−0.525** | **−0.546** | 0.442 | 0.362 |
| miR-516b-5p # | 340 | **−0.512** | 0.418 | −0.406 | −0.411 | 0.444 | 0.403 |
| miR-519d-3p | 329 | **−0.611** | **0.653** | **−0.556** | −0.433 | **0.529** | 0.488 |
| miR-518b | 200 | **−0.568** | **0.571** | **−0.618** | −0.446 | 0.415 | 0.388 |
| miR-522-3p * | 164 | **−0.581** | **0.529** | −0.490 | −0.468 | **0.503** | 0.453 |
| miR-526b-5p #* | 162 | **−0.547** | **0.598** | −0.401 | −0.459 | **0.584** | 0.418 |
| miR-515-5p #* | 150 | **−0.553** | **0.532** | −0.417 | −0.406 | **0.524** | 0.497 |
| miR-520g-3p #* | 142 | **−0.673** | **0.777** | **−0.593** | −0.491 | **0.541** | 0.391 |
| miR-518e-3p | 135 | **−0.561** | **0.506** | **−0.530** | −0.433 | 0.459 | 0.435 |
| miR-518f-3p | 87 | **−0.749** | **0.703** | **−0.548** | **−0.639** | 0.365 | 0.356 |
| miR-518f-5p * | 77 | −0.447 | 0.374 | −0.257 | −0.364 | **0.506** | 0.482 |
| miR-518a-3p | 59 | **−0.642** | **0.600** | **−0.582** | **−0.518** | 0.453 | 0.441 |
| miR-525-5p #* | 52 | **−0.595** | 0.485 | −0.418 | **−0.509** | 0.338 | 0.324 |
| miR-517-5p * | 46 | **−0.595** | **0.568** | **−0.522** | −0.471 | 0.482 | 0.453 |
| miR-520f-3p | 36 | **−0.506** | **0.576** | −0.238 | −0.362 | **0.625** | 0.482 |
| miR-520d-5p #* | 26 | −0.316 | 0.212 | −0.093 | −0.358 | 0.488 | **0.500** |
| miR-520d-3p | 26 | **−0.567** | 0.491 | **−0.515** | −0.471 | 0.491 | 0.459 |
| miR-512-3p #* | 25 | **−0.587** | 0.488 | **−0.569** | **−0.499** | 0.315 | 0.253 |
| miR-498-5p | 23 | **−0.586** | 0.491 | −0.420 | **−0.506** | 0.385 | 0.353 |
| miR-518c-5p | 18 | −0.380 | 0.426 | −0.211 | −0.320 | **0.604** | **0.509** |
| miR-520h #* | 18 | **−0.723** | **0.700** | **−0.641** | **−0.608** | 0.335 | 0.306 |
| miR-519d-5p #* | 12 | **−0.589** | 0.591 | **−0.533** | −0.434 | 0.512 | 0.479 |
| miR-519c-3p | 11 | **−0.605** | 0.521 | **−0.535** | −0.496 | 0.438 | 0.429 |
| miR-525-3p | 11 | **−0.587** | 0.585 | **−0.609** | −0.415 | 0.397 | 0.362 |

The correlation coefficient, R, is given. Significant correlations ($P < 0.05$) are shown in bold. Only miRNAs with a mean signal > 10 CPM were included in the analysis. Depending on the normal distribution of data, either Pearson's or Spearman's correlation was used. #Eleven miRNAs targeting the insulin receptor (*INSR*) mRNA. *Fourteen miRNAs targeting insulin receptor substrate 1 and/or 2 (*IRS1/2*) mRNAs. Target analysis was performed using the TargetScan online tool. Abbreviations: CPM, counts per million; GWG, gestational weight gain; WR, weight ratio.

## Pregnancy-specific miRNA clusters

Given that cluster analysis revealed a remarkable correlation among miRNAs of the C19MC and maternal and neonatal characteristics, these miRNAs were analysed in more detail. Among the C19MC miRNAs, 33 showed significant correlations with parameters such as placental weight (25 miRNAs), fetoplacental weight ratio (21 miRNAs), GWG (15 miRNAs), UCP insulin (14 miRNAs), neonatal weight (9 miRNAs) or neonatal insulin resistance (HOMA-IR; 5 miRNAs) (Table 2). Given the associations with UCP insulin and neonatal HOMA-IR, we analysed the mRNA targets of the C19MC

miRNAs to explore potential links to insulin signalling. Using the online tool TargetScan, we identified that 11 of the 33 miRNAs target insulin receptor (*INSR*) mRNA and 14 target insulin receptor substrate 1 or 2 (*IRS1/2*) mRNAs (Table 2).

In contrast, miRNAs from the C14MC did not show such consistent associations or clustering patterns. Nevertheless, 35 mature miRNAs from the C14MC were significantly correlated with maternal and/or neonatal characteristics. These miRNAs showed positive associations with maternal pre-pregnancy BMI and BMI at birth and with UCP leptin, as indicated by red colouring in the heatmaps in Fig. 2*A* and *B*. Specifically, 13 C14MC

**Table 3. Correlation of pre-pregnancy body mass index, body mass index at birth and umbilical cord plasma leptin with chromosome 14 miRNA cluster microRNAs in umbilical cord plasma**

| MicroRNA ID | Mean CPM | Pre-pregnancy BMI | | BMI at birth | | Leptin | |
|---|---|---|---|---|---|---|---|
| | | *R* | *P*-value | *R* | *P*-value | *R* | *P*-value |
| miR-376c-3p | 2288 | **0.509** | **0.044** | **0.642** | **0.010** | 0.478 | 0.061 |
| miR-411-5p | 1478 | 0.464 | 0.070 | **0.539** | **0.038** | 0.183 | 0.497 |
| miR-369-5p | 1035 | **0.499** | **0.049** | **0.615** | **0.015** | 0.081 | 0.765 |
| miR-431-5p | 1017 | **0.592** | **0.016** | **0.587** | **0.021** | 0.320 | 0.227 |
| miR-379-5p | 459 | 0.365 | 0.164 | **0.590** | **0.021** | 0.173 | 0.523 |
| miR-337-5p | 345 | 0.409 | 0.115 | **0.617** | **0.014** | 0.225 | 0.403 |
| miR-487b-3p | 229 | **0.608** | **0.012** | 0.129 | 0.648 | **0.500** | **0.049** |
| miR-493-5p | 224 | **0.758** | **0.001** | 0.421 | 0.118 | **0.499** | **0.049** |
| miR-369-3p | 158 | 0.405 | 0.120 | 0.500 | 0.057 | **0.524** | **0.037** |
| miR-409-5p | 59 | **0.672** | **0.004** | **0.690** | **0.004** | 0.423 | 0.102 |
| miR-154-3p | 49 | **0.503** | **0.047** | **0.554** | **0.032** | **0.656** | **0.006** |
| miR-329-3p | 45 | 0.456 | 0.076 | 0.460 | 0.085 | **0.623** | **0.010** |
| miR-299-5p | 39 | **0.519** | **0.039** | 0.298 | 0.280 | 0.306 | 0.250 |
| miR-496 | 32 | 0.481 | 0.060 | 0.464 | 0.082 | **0.524** | **0.037** |
| miR-495-3p | 30 | 0.462 | 0.072 | 0.338 | 0.217 | **0.571** | **0.021** |
| miR-433-3p | 26 | **0.550** | **0.027** | 0.498 | 0.059 | **0.594** | **0.015** |
| miR-370-3p | 26 | 0.368 | 0.161 | 0.343 | 0.211 | **0.538** | **0.031** |
| miR-655-3p | 23 | 0.441 | 0.087 | 0.432 | 0.108 | **0.574** | **0.020** |
| miR-379-3p | 22 | **0.574** | **0.020** | **0.629** | **0.012** | **0.600** | **0.014** |
| miR-485-3p | 20 | **0.522** | **0.038** | 0.345 | 0.208 | **0.522** | **0.038** |
| miR-377-3p | 18 | 0.199 | 0.459 | 0.164 | 0.560 | **0.563** | **0.023** |
| miR-376a-5p | 15 | **0.554** | **0.026** | 0.364 | 0.183 | **0.528** | **0.035** |
| miR-543 | 11 | **0.506** | **0.046** | **0.545** | **0.036** | **0.535** | **0.033** |
| miR-758-3p | 11 | 0.462 | 0.072 | 0.357 | 0.191 | **0.576** | **0.019** |

The correlation coefficient, *R*, and the *P*-value are given. Significant correlations ($P < 0.05$) are shown in bold. Only microRNAs with a mean signal of >10 CPM were included in the analysis. Depending on the normal distribution of data, either Pearson's or Spearman's correlation was used. Abbreviations: CPM, counts per million; BMI, body mass index.

miRNAs were significantly correlated with pre-pregnancy BMI, 10 with maternal BMI at birth and 16 with UCP leptin. The correlations between C14MC miRNAs and maternal/neonatal characteristics are summarized in Table 3 and visualized in a heatmap in Fig. 3*A*, which includes only characteristics correlated with more than one C14MC mRNA. Graphs depicting the miRNAs with strongest signals and correlating with maternal BMI, both before pregnancy and at delivery, are shown in Fig. 3*B–G*. Additionally, miRNAs correlated with pre-pregnancy BMI or BMI at birth were subjected to pathway analysis using miRPath online tool. The top 10 significant pathways identified are listed in Table 4.

### Distinct UCP miRNA signatures in male *vs.* female newborns

Fetal sex is a central determinant of neonatal characteristics, including anthropometrics and levels of hormones and growth factors (Lehre et al., 2013; Stern et al., 2021). Therefore, we investigated whether UCP miRNAs differ between male and female newborns. Subject characteristics of pregnancies with male *vs.* female newborns are shown in Table 5. Among these characteristics, leptin levels were higher in UCP of female newborns, whereas all other parameters were comparable between the two groups. Point-biserial correlation analysis identified 21 miRNAs with significantly different levels between male and female neonates (Table 6), with the majority showing higher expression in the male neonates. Notably, only two miRNAs, miR-424-5p and miR-513c-5p, were encoded by sex chromosomes. Table 6 also lists maternal or neonatal parameters with which the respective miRNAs were correlated. None of the miRNAs that differed between male and female neonates was correlated with maternal BMI (pre-pregnancy or at birth) or with UCP leptin, despite the latter being elevated in female neonates. Pathway analysis of miRNAs with sex differences using miRPath 4.0 online tool based on TarBase and TargetScan revealed 'cell cycle', 'p53 signalling pathway', and 'ubiquitin mediated proteolysis' as the top three pathways.

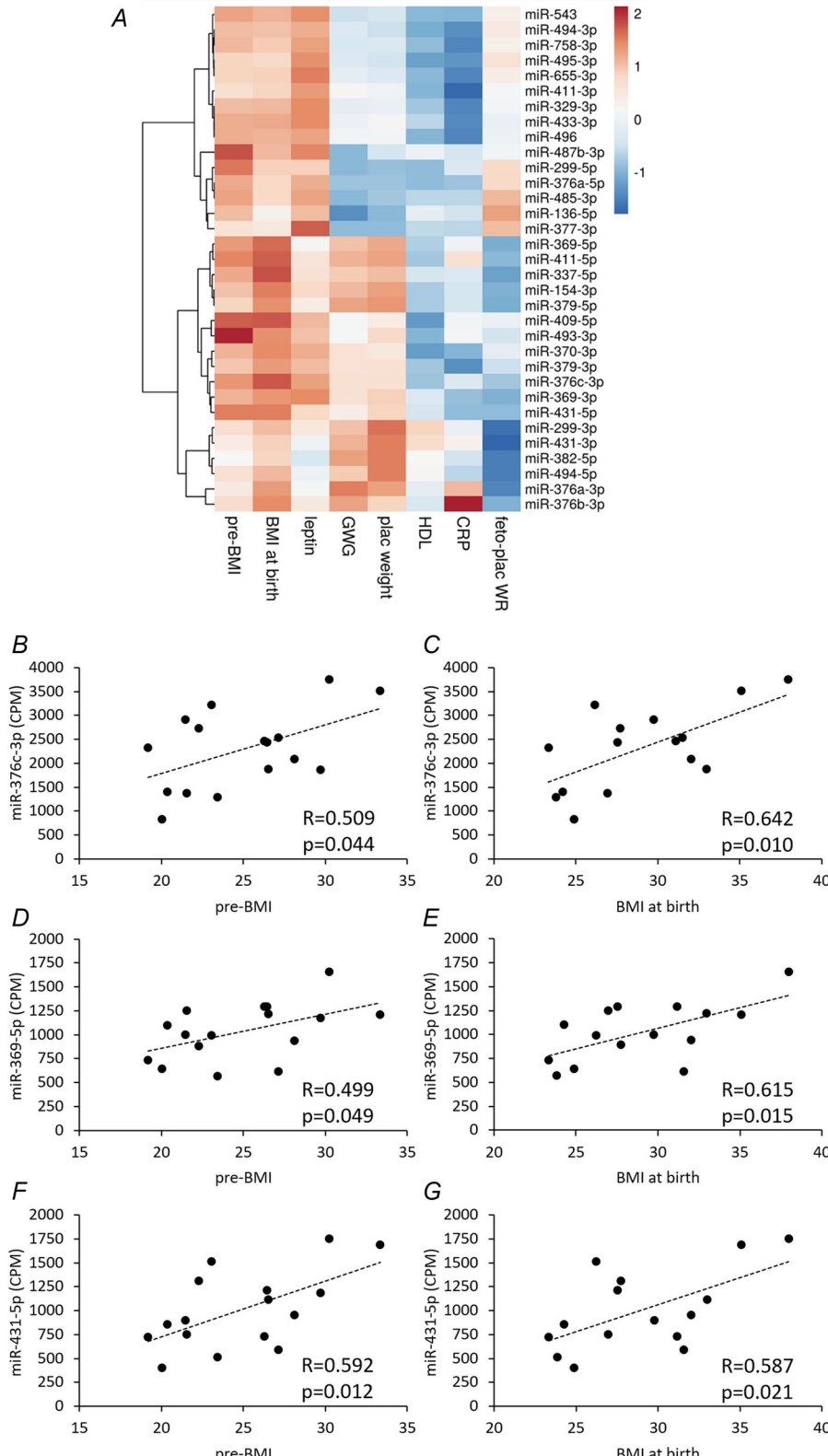

**Figure 3. Correlation of chromosome 14 miRNA cluster microRNAs in umbilical cord plasma with maternal and neonatal characteristics**

*A*, heatmap illustrating the correlations of C14MC miRNAs with maternal and neonatal characteristics. Only miRNAs with a mean signal of >10 CPM and characteristics that correlated with more than one C14MC miRNA were included. Correlation coefficients were used to generate the heatmap and to cluster the miRNAs (on the

left) using correlation distance clustering and McQuitty linkage, as implemented in the ClustVis online tool. *B–G*, correlations of three selected C14MC miRNAs (miR-376c-3p, miR-369-5p and miR-431-5p) are shown, because these miRNAs had the highest signals and correlated with both pre-pregnancy body mass index and body mass index at birth. Depending on the normal distribution of data, Pearson's or Spearman's correlation analyses were performed. The correlation coefficient (*R*) and *P*-value are indicated in the plots. Abbreviations: C14MC, chromosome 14 miRNA cluster; CPM, counts per million; feto-plac WR, fetoplacental weight ratio; GWG, gestational weight gain; miRNA, microRNA; neo length, neonatal length; neo weight, neonatal weight; plac weight, placental weight; Pre-BMI, pre-pregnancy BMI; TG, triglycerides. [Colour figure can be viewed at wileyonlinelibrary.com]

**Table 4. Pathway analysis of chromosome 14 miRNA cluster microRNAs correlating with maternal pre-pregnancy body mass index and/or body mass index at birth**

| Pathway | Total genes in PW | Target genes in PW | *P*-value after FDR |
|---|---|---|---|
| Focal adhesion | 213 | 46 | <0.000000004 |
| PI3K-Akt signalling pathway | 372 | 55 | <0.00002 |
| Hippo signalling pathway | 164 | 32 | <0.00002 |
| Ubiquitin-mediated proteolysis | 142 | 29 | <0.00002 |
| FoxO signalling pathway | 139 | 27 | <0.00009 |
| TGF-beta signalling pathway | 103 | 22 | 0.0001 |
| EGFR tyrosine kinase inhibitor resistance | 82 | 19 | 0.0002 |
| Adherens junction | 79 | 18 | 0.0003 |
| Rap1 signalling pathway | 214 | 33 | 0.0009 |
| Protein processing in endoplasmic reticulum | 194 | 30 | 0.002 |

Pathway analysis used miRPath v.4.0 based on TarBase and TargetScan. Only the top 10 pathways are shown. Cancer-related pathways were removed from the output list. Abbreviations: FDR, false discovery rate; PW, pathway.

## Discussion

Umbilical cord plasma miRNAs are an emerging field of research, because they not only serve as biomarkers, but also represent biologically active molecules with regulatory roles in both physiology and pathology. In our study, we adopted a global miRNA-specific next-generation sequencing approach to investigate the associations of UCP miRNAs with maternal metabolic parameters and neonatal anthropometric, metabolic and inflammatory characteristics. Our primary findings reveal that a majority of UCP miRNAs are sensitive to maternal and neonatal characteristics. Most notably, they are associated with pre-pregnancy BMI, BMI at birth and GWG, in addition to placental weight, fetoplacental weight ratio, UCP leptin and UCP CRP. Furthermore, we observed a strong association between placenta-derived C19MC miRNAs and placental weight and UCP insulin neonatal and insulin resistance, in addition to between pregnancy-specific C14MC miRNAs with maternal BMI and UCP leptin.

In our study, 31.8% of the 387 miRNAs detected above a threshold of 10 CPM were correlated with maternal weight-related measures, i.e. maternal pre-pregnancy BMI, maternal BMI at birth and GWG. This finding aligns with previous studies demonstrating an interaction between maternal BMI, GWG and cell-free circulating miRNAs in umbilical cord blood (Juracek et al., 2019;

Mendez-Mancilla et al., 2018). Using miRNA-specific arrays to analyse UCP of a cohort of 19 women with normal BMI status (BMI 18.5–25 kg/m$^2$) and 5 women with overweight (BMI > 25 kg/m$^2$), Juracek et al. (2019) identified 14 miRNAs associated with maternal BMI and 9 miRNAs associated with GWG. By measuring four candidate miRNAs in umbilical cord serum from 21 normal-weight and 20 overweight/obese women using RT-qPCR, Mendez-Mancilla et al. (2018) identified three miRNAs at different levels. Remarkably, all three, i.e. miR-155, miR-181a and miR-221, were correlated with either pre-pregnancy BMI or GWG in our study.

In the present study, 14.0% of the miRNAs detected above a threshold of 10 CPM were associated with maternal glycaemia at mid-pregnancy. Although maternal glycaemia plays a crucial role in fetal development, the relationship between maternal glycaemia and UCP miRNAs has not been investigated previously. Given that all participants in our study were healthy and not diagnosed with gestational diabetes, these results highlight that even subtle metabolic differences within the normal, healthy range can influence the fetal epigenome.

Regarding the effects of neonatal traits on the UCP miRNA profile, the largest group of miRNAs was associated with the fetoplacental weight ratio (15.5%), followed by UCP leptin (14.5%) and placental weight (13.7%). To our knowledge, such associations have not been reported before. The strong association between

**Table 5. Characteristics of pregnancies with male *vs*. female neonates**

| Characteristic | Male | Female | *P*-value |
|---|---|---|---|
| Sample size | 8 | 8 | |
| Mode of delivery (vaginal/caesarean section) | 3/5 | 6/2 | |
| Pregnancy length (weeks) | $39.6 \pm 0.9$ | $39.1 \pm 0.9$ | 0.251 |
| Maternal parameters | | | |
| Maternal age (years) | $29.6 \pm 6.0$ | $29.3 \pm 3.5$ | 0.882 |
| Pre-pregnancy BMI (kg/m$^2$) | $24.3 \pm 4.1$ | $25.6 \pm 4.4$ | 0.525 |
| BMI at birth (kg/m$^2$) | $29.3 \pm 4.6$ | $28.7 \pm 4.5$ | 0.800 |
| Gestational weight gain (kg) | $14.4 \pm 7.2$ | $9.5 \pm 5.8$ | 0.172 |
| oGTT 0 h (mg/dl) | $78.9 \pm 9.0$ | $79.8 \pm 7.8$ | 0.745 |
| oGTT 1 h (mg/dl) | $116.0 \pm 31.4$ | $128.0 \pm 24.2$ | 0.406 |
| oGTT 2 h (mg/dl) | $100.1 \pm 30.3$ | $113.4 \pm 18.9$ | 0.322 |
| Neonatal parameters | | | |
| Placental weight (g) | $673 \pm 133$ | $591 \pm 149$ | 0.269 |
| Fetoplacental weight ratio | $5.29 \pm 0.56$ | $5.72 \pm 0.92$ | 0.284 |
| Neonatal weight (g) | $3506 \pm 429$ | $3274 \pm 330$ | 0.246 |
| Neonatal length (cm) | $51.4 \pm 3.2$ | $49.6 \pm 2.2$ | 0.280 |
| Umbilical cord plasma parameters | | | |
| Glucose (mg/dl) | $67.1 \pm 14.5$ | $71.9 \pm 16.4$ | 0.547 |
| C-peptide (pmol/l) | $235 \pm 68$ | $246 \pm 73$ | 0.772 |
| Insulin (μU/ml) | $8.9 \pm 5.7$ | $10.9 \pm 6.1$ | 0.513 |
| HOMA-IR | $1.54 \pm 1.10$ | $2.05 \pm 1.72$ | 0.645 |
| CRP (ng/ml) | $90.0 \pm 22.0$ | $75.0 \pm 18.5$ | 0.163 |
| IL6 (pg/ml) | $8.87 \pm 3.82$ | $6.95 \pm 4.00$ | 0.266 |
| TNF$\alpha$ (pg/ml) | $0.957 \pm 0.166$ | $0.872 \pm 0.145$ | 0.306 |
| hGH (μg/ml) | $13.4 \pm 7.9$ | $11.5 \pm 8.5$ | 0.721 |
| Leptin (ng/ml) | $1.51 \pm 1.02$ | $3.06 \pm 1.53$ | **0.032** |
| HDL (mmol/l) | $0.73 \pm 0.19$ | $0.69 \pm 0.13$ | 0.696 |
| LDL (mmol/l) | $0.57 \pm 0.30$ | $0.47 \pm 0.17$ | 0.444 |
| Cholesterol (mg/dl) | $59.2 \pm 20.9$ | $54.9 \pm 8.1$ | 0.595 |
| Triglycerides (mg/dl) | $35.1 \pm 10.6$ | $31.5 \pm 14.9$ | 0.603 |

Data are presented as the mean $\pm$ SD. *P*-values < 0.05 are printed in bold. Abbreviation: BMI, body mass index; CRP, c-reactive protein; HDL, high-density lipoprotein; hGH, human growth hormone; HOMA-IR, homeostatic model assessment for insulin resistance; IL6, interleukin 6; LDL, low-density lipoprotein; oGTT, oral glucose tolerance test; TNF$\alpha$, tumor necrosis factor alpha.

placental weight and UCP miRNAs might be attributable to the fact that the placenta is a source of many fetal circulating miRNAs, including C19MC miRNAs (Paquette et al., 2018).

Our cluster analysis based on the correlation between UCP miRNAs and maternal and neonatal characteristics revealed a consistant pattern among C19MC miRNAs. These miRNAs exhibited negative correlations with placental weight and positive correlations with UCP insulin and insulin resistance (HOMA-IR). MicroRNA clusters, such as C19MC, represent genomic regions encoding groups of miRNAs in close proximity, which are often co-regulated and co-expressed. The C19MC, unique to primates, is pregnancy specific and expressed primarily in the placental trophoblast (Paquette et al., 2018). Indeed, C19MC miRNA expression has been strongly linked to placental development and function (Hromadnikova et al., 2015, Xie & Sadovsky, 2016). The

C19MC miRNAs are released from the placenta into the maternal circulation (Morales-Prieto et al., 2013), representing a pathway for fetomaternal communication (Morales-Prieto et al., 2020). Specifically, C19MC miRNAs have been implicated in maternal adaptation to pregnancy, because they are highly abundant in maternal plasma during gestation and have been shown to be predictive of mid-pregnancy insulin sensitivity (Legare et al., 2022) and fasting glucose levels (Legare et al., 2024) as early as in the first trimester. Furthermore, C19MC miRNAs might support the maternal immune system (Chaiwangyen et al., 2020) during viral infections (Dumont et al., 2017).

Notably, the placenta also releases C19MC miRNAs into the fetal circulation at levels comparable to those found in the maternal circulation (Paquette et al., 2018), suggesting a role for C19MC miRNAs in the fetal compartment as well. However, the function of C19MC

**Table 6. MicroRNAs with different levels in umbilical cord plasma of male *vs*. female neonates**

| MicroRNA ID | Female | Male | Fold change | P-value | Chromosome | Correlating parameters |
|---|---|---|---|---|---|---|
| miR-30e-5p | 16,381 | 19,319 | 1.18 | 0.041 | 1 | Maternal: GWG |
| | | | | | | Neonatal: fp WR, TNFA |
| miR-185-5p | 13,583 | 17,168 | 1.26 | 0.010 | 22 | |
| miR-130a-3p | 8183 | 12,575 | 1.54 | 0.021 | 11 | Maternal: oGTT 1 h |
| | | | | | | Neonatal: TNFA, LDL |
| miR-15a-5p | 903 | 1317 | 1.46 | 0.004 | 13 | Maternal: oGTT 2 h |
| | | | | | | Neonatal: LDL |
| miR-424-5p | 760 | 1252 | 1.65 | 0.041 | X | |
| miR-107 | 501 | 642 | 1.28 | 0.004 | 10 | Neonatal: insulin, HOMA-IR, glucose |
| miR-29a-3p | 415 | 687 | 1.66 | 0.041 | 7 | Neonatal: CRP, HDL, IL6, cholesterol, LDL |
| miR-376b-3p | 399 | 683 | 1.71 | 0.001 | 14 | Neonatal: CRP |
| miR-210-3p | 352 | 601 | 1.71 | 0.041 | 11 | Neonatal: CRP |
| miR-101-3p | 224 | 345 | 1.54 | 0.041 | 1 | Neonatal: fp WR, IL6 |
| miR-29c-3p | 145 | 278 | 1.92 | 0.001 | 1 | Neonatal: LDL |
| miR-598-3p | 90 | 108 | 1.20 | 0.021 | 8 | – |
| miR-345-5p | 66 | 91 | 1.38 | 0.041 | 14 | – |
| miR-589-5p | 40 | 57 | 1.44 | 0.041 | 7 | Neonatal: TNFA, LDL |
| miR-4508 | 57 | 31 | 0.55 | 0.030 | 15 | – |
| miR-627-5p | 24 | 52 | 2.18 | 0.030 | 15 | Neonatal: cholesterol |
| miR-3611 | 20 | 32 | 1.65 | 0.030 | 10 | Maternal: GWG, oGTT 1 h |
| | | | | | | Neonatal: fp WR, cholesterol, TNFA, LDL |
| miR-513c-5p | 7 | 37 | 5.49 | 0.004 | X | Maternal: oGTT 1 h |
| | | | | | | Neonatal: hGH, IL6 |
| miR-499a-5p | 16 | 25 | 1.57 | 0.030 | 20 | Maternal: oGTT 1 h, oGTT 2 h |
| miR-519b-3p | 29 | 11 | 0.37 | 0.041 | 19 | – |
| miR-590-5p | 10 | 16 | 1.65 | 0.000 | 7 | Maternal: GWG |
| | | | | | | Neonatal: fp WR |

Signals [in counts per million (CPM)] of microRNAs in umbilical cord blood plasma of male and female neonates (*n* = 8/8) are presented as the mean. The fold change is between male and female. The chromosome number of the chromosome encoding the respective miRNA. The column correlating parameters lists maternal or neonatal characteristics correlating with the respective microRNA. Statistical analysis used point-biserial correlation.

miRNAs in the fetal circulation remains less understood compared with their role in the maternal circulation. In whole umbilical cord blood containing blood cells, a positive association between certain C19MC miRNAs and maternal pre-gestational BMI has been identified (Jing et al., 2020). However, the specific umbilical cord blood cell type contributing to the detected C19MC miRNAs is unknown. In our study, which focused on UCP, no such association between C19MC miRNAs and maternal BMI was observed. Interestingly, we identified a negative correlation between C19MC miRNAs and placental weight. This appears to contrast with findings in mice, where overexpression of human C19MC resulted in larger placentas (Mouillet et al., 2020). However, overexpression in mice might lead to the release of super- or supraphysiological levels of placental miRNAs, whereas in humans the release of placental miRNAs into the maternal or fetal circulation is a regulated process, with distinct patterns of individual miRNAs observed between the

mother, fetus and placenta (Chang et al., 2017; Paquette et al., 2018). Consequently, the levels of C19MC miRNAs detected in UCP might differ from those present within the placenta itself.

We can only speculate about the function of C19MC in UCP. As a hypothesis, we propose that low placental weight, which is often associated with low fetal weight (Sanin et al., 2001), might result in an increased release of C19MC miRNA signatures from the placenta into the fetal circulation. In parallel to the aforementioned association of C19MC miRNAs in the maternal circulation with maternal insulin sensitivity (Legare et al., 2022), several of the C19MC miRNAs in UCP were correlated with UCP insulin and neonatal HOMA-IR in our study. Nine of these C19MC miRNAs target mRNAs encoding the insulin receptor and/or insulin receptor substrates 1 and 2. These findings suggest that higher levels of C19MC miRNAs in UCP might contribute to fetal insulin resistance, a condition associated with increased fetal body fat

(Catalano et al., 2009). Consequently, increased release of C19MC miRNAs by smaller placentas might play a role in inducing fetal insulin resistance and, thereby, promoting fetal growth. Notably, among maternal characteristics, only GWG was correlated with a higher number of C19MC miRNAs. In fact, GWG and placental weight are strongly correlated, as observed previously (Zhang et al., 2023), and this relationship was also evident in our cohort ($R = 0.855$; $P = 0.0005$).

Another pregnancy-specific miRNA cluster, C14MC, differs from C19MC in that it is not placenta specific (Morales-Prieto et al., 2020). The function of C14MC remains less clear than that of C19MC. However, its expression patterns during pregnancy suggest a role in embryonic, fetal and placental development (Dini et al., 2019). Our data revealed higher C14MC miRNA levels in the fetal circulation of mothers with higher BMIs. Pathway analysis of the C14MC miRNAs correlated with pre-pregnancy BMI or BMI at birth in our study assigned these miRNAs to 'focal adhesion' and 'adherens junction', indicating an association with endothelial barrier function of the placenta and fetus. In fact, previous research has identified a role for C14MC miRNAs in endothelial function (Dini et al., 2018).

Several neonatal characteristics, including anthropometric measures and concentrations of hormones and growth factors, show sexual dimorphism (Lehre et al., 2013; Stern et al., 2021). To investigate whether UCP miRNAs also display sex-specific differences, we analysed their levels in male and female neonates. Our study identified 21 circulating miRNAs with significantly different levels between the sexes. In a population-based cohort study investigating plasma miRNAs in 372 adults aged 22–79 years, Ameling et al. (2015) identified 35 sex-specific miRNAs, of which 5 miRNAs (miR-130a-3p, miR-15a-5p, miR-424-5p, miR-101-3p and miR-29c-3p) overlapped with the 21 miRNAs showing sexual dimorphism in our study. This highlights that certain sex-specific plasma miRNAs are independent of age and are present even before puberty. Investigating the presence of these miRNAs during childhood and close to puberty could provide valuable insight into the potential sex-specific roles. Although sexual dimorphism is evident in placental hormones (Stern et al., 2021) and neonatal fat mass (Rodriguez et al., 2004; Shields et al., 2006), we did not observe sex differences in any miRNAs associated with maternal BMI. This suggests that the associations of miRNAs with maternal and neonatal metabolism, growth factors and inflammatory traits are largely independent on fetal sex. Pathway analysis suggested that miRNAs with sex differences might be related to growth and tissue turnover. In fact, studies based on larger cohorts revealed higher birth weight of male neonates (Voskamp et al., 2020), which might be reflected by the UCP miRNA profile.

Circulating miRNAs are regulatory molecules with the potential to affect cells or tissues functionally or structurally. This raises the question of whether altered miRNA profiles at birth might predict or even actively influence the future development of the newborn. A study by Takatani et al. (2022) addressed this question. Based on BMI at 1.5, 3 and 5 years of age, children participating in a birth cohort study were classified into high and low obesity risk groups. Stored umbilical cord serum samples from five children per group were analysed using miRNA-specific arrays, and the results were validated by RT-PCR in 33 umbilical cord serum samples per group. The study identified five miRNAs that were altered in umbilical cord serum of newborns at higher risk of becoming obese later in life. Among these were also miR-130a-3p, which in our cohort was correlated with maternal oGTT levels at 1 h and with UCP LDL and TNF$\alpha$, and miR-1260b, which was correlated with maternal pre-pregnancy BMI and UCP leptin. Interestingly, miR-130a-3p has also been shown to influence adipocyte differentiation and function, highlighting its potential role in metabolic programming (Greither et al., 2020; Wu et al., 2020).

We consider the detailed characterization of our cohort, especially the UCP, to be a major strength of this study. However, we acknowledge that a larger sample size would allow for more detailed analysis and correlations. In addition, conducting a more comprehensive metabolic profile of the mothers, similar to that performed for the newborns, including assessment of maternal insulin resistance and leptin levels, would provide valuable insights. Given that C19MC is associated with fetal insulin and insulin resistance, additional information on maternal insulin resistance would be of particular interest. Finally, we are aware that many circulating miRNAs are transported in extracellular vesicles, which, through their surface molecules, determine the target cells they influence (Czernek & Duchler, 2020). These exosomes participate in fetomaternal communication (Adam et al., 2017; Chaiwangyen et al., 2020; Donker et al., 2012). Although we analysed the entirety of circulating miRNAs, it would be highly interesting to investigate the subset of miRNAs transported in extracellular vesicles.

In conclusion, our study highlights the sensitivity of circulating neonatal miRNAs to maternal metabolic conditions, demonstrates the associations of UCP miRNAs with neonatal metabolic and inflammatory traits, and underscores the potential role of circulating umbilical cord blood miRNAs in fetal metabolism and development.

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

## Additional information

### Data availability statement

The next-generation sequencing data from this study are deposited at the Gene Expression Omnibus (GEO) database (www.ncbi.nlm.nih.gov/geo) under record no. GSE276266.

### Competing interests

The authors have no conflict of interest.

### Author contributions

J.Z.-S. and U.H.: conception or design of the work; J.Z.-S., A.R., C.T., M.S.D., E.W. and U.H.: acquisition, analysis or interpretation of data for the work; J.Z.-S. and U.H.: drafting the work or revising it critically for important intellectual content;

J.Z.-S., A.R., C.T., M.S.D., E.W. and U.H.: final approval of the version to be published; J.Z.-S., A.R., C.T., M.S.D., E.W. and U.H.: agreement to be accountable for all aspects of the work in ensuring that questions related to the accuracy or integrity of any part of the work are appropriately investigated and resolved. All persons designated as authors qualify for authorship, and all those who qualify for authorship are listed.

## Funding

This research was funded in whole or in part by the Austrian Science Fund (FWF) [10.55776/KLI1023]. This study was further supported by the ZFF_Impuls_22 funding from Carinthia University of Applied Sciences (CUAS), project title "MIPROPREG", project number PJ02. For open access purposes, the author has applied a CC BY public copyright license to any author accepted manuscript version arising from this submission.

## Keywords

circulating microRNAs, maternal body mass index, neonatal homeostatic model assessment of insulin resistance, placental weight, umbilical cord plasma

## Supporting information

Additional supporting information can be found online in the Supporting Information section at the end of the HTML view of the article. Supporting information files available:

**Peer Review History**
**Supplementary Table 1**

