## [Peer Review History · The Journal of Physiology]

MiRNA Profiling in Umbilical Cord Plasma: Links to Maternal Metabolism and Neonatal Metabolic and Inflammatory Traits

Jasmin Strutz, Anna Rieder, Carolina Tocantins, Mariana S. Diniz, Elisa Weiss, and Ursula Hiden

DOI: 10.1113/JP287672

Corresponding author(s): Ursula Hiden (ursula.hiden@medunigraz.at)

The following individual(s) involved in review of this submission have agreed to reveal their identity: Ashley S Meakin (Referee #1); Kasia Maksym (Referee #2)

Review Timeline:

Submission Date:	10-Sep-2024
Editorial Decision:	17-Oct-2024
Revision Received:	08-Jan-2025
Accepted:	05-Feb-2025

Senior Editor: Laura Bennet

Reviewing Editor: Janna Morrison

Transaction Report:

Dear Dr Hiden,

Re: JP-RP-2024-287672 "MiRNA Profiling in Umbilical Cord Plasma: Links to Maternal Metabolism and Neonatal Metabolic and Inflammatory Traits" by Jasmin Strutz, Anna Rieder, Carolina Tocantins, Mariana S. Diniz, Elisa Weiss, and Ursula Hiden

Thank you for submitting your manuscript to The Journal of Physiology. It has been assessed by a Reviewing Editor and by 2 expert referees and we are pleased to tell you that it is potentially acceptable for publication following satisfactory major revision.

LANGUAGE EDITING AND SUPPORT FOR PUBLICATION: If you would like help with English language editing, or other article preparation support, Wiley Editing Services offers expert help, including English Language Editing, as well as translation, manuscript formatting, and figure formatting at www.wileyauthors.com/eoo/preparation. You can also find resources for Preparing Your Article for general guidance about writing and preparing your manuscript at www.wileyauthors.com/eoo/prepresources.

REVISION CHECKLIST:

We look forward to receiving your revised submission.

Yours sincerely,

Laura Bennet
Senior Editor
The Journal of Physiology

REQUIRED ITEMS

- Author photo and profile. First or joint first authors are asked to provide a short biography (no more than 100 words for one author or 150 words in total for joint first authors) and a portrait photograph. These should be uploaded and clearly labelled together in a Word document with the revised version of the manuscript. See Information for Authors for further details.
- You must start the Methods section with a paragraph headed Ethical Approval. If experiments were conducted on humans, confirmation that informed consent was obtained, preferably in writing, that the studies conformed to the standards set by the latest revision of the Declaration of Helsinki and that the procedures were approved by a properly constituted ethics committee, which should be named, must be included in the article file. If the research study was registered (clause 35 of the Declaration of Helsinki), the registration database should be indicated, otherwise the lack of registration should be noted as an exception (e.g. The study conformed to the standards set by the Declaration of Helsinki, except for registration in a database). For further information see: <https://physoc.onlinelibrary.wiley.com/hub/human-experiments>.
- The reference list must be in alphabetical order, rather than numbered, to comply with our Journal format.
- Your manuscript must include a complete Additional Information section, including competing interests; funding; author contributions and acknowledgements.
- The Journal of Physiology funds authors of provisionally accepted papers to use the premium BioRender site to create high resolution schematic figures. Follow this link and enter your details and the manuscript number to create and download figures. Upload these as the figure files for your revised submission. If you choose not to take up this offer, we require figures to be of similar quality and resolution. If you are opting out of this service to authors, state this in the Comments section on the Detailed Information page of the submission form. The link provided should only be used for the purposes of this submission. Authors will be charged for figures created on this premium BioRender account if they are not related to this manuscript submission.
- Please upload separate high-quality figure files via the submission form.

Reviewing Editor's comments:

Graphical abstract - Should the text at the bottom of the figure use the terms 'responsive to' rather than 'sensitive towards'?

The introduction is very brief. The rationale of the role of miR in relation to maternal BMI could be expanded.

Line 89 - UCB was collected from healthy pregnancies. Were a range of maternal BMIs included? How was this determined as only 16 samples were collected? Was there an a priori decision to collect from an equal number of male and female pregnancies? How many placentas were collected at each BMI?

Was higher maternal age associated with higher maternal BMI? If yes, may this impact the findings?

Referee #1:

Overall, the work by Strutz et al., reads well and provides insight into UCP miRNAs and their association with maternal and fetal morphometric measures as well as some markers of metabolic health.

Introduction

The introduction is very succinct; however, based on the layout of Table 1, it would make sense to expand on how fetoplacental responses are different between sexes in normal and overweight pregnancies.

Methods

Were any of the maternal or neonatal factors included as covariates in the miRNA analysis pipeline?

Results

Table 1. Can the authors comment on the rationale for displaying the data split by sex, if the focus of the study was to analyse UCP miRNAs in relation to maternal BMI? It would make more sense to compare outcomes by BMI throughout the study and, where appropriate, stratify by sex.

UCP miRNA profiling. The authors state that they focussed on miRNAs with a mean CPM >100 - what was the rationale for using this cut off? Considering many (~90%) of miRNAs had a CPM <100, is it likely that biologically meaningful findings have inadvertently been overlooked based on these parameters?

'Distinct UCP miRNA signatures in male vs. female newborns' - are these signatures unique to normal or overweight BMI pregnancies? Have similar analyses been performed between male <25 vs >25 BMI and female <25 vs >25 BMI

pregnancies? This would certainly be interesting to examine.

Discussion

The sex-specific differences section of the discussion could expand to include any known role of these miRNAs in the regulation of genes required for placental development/fetal growth.

Referee #2:

Thank you for submitting your paper.

The results presented in the article is opening new area of research and contributes to understanding of the physiology and pathology of interaction between fetus, placenta and mother. The analysis of miRNA cast light on molecular interactions and impact of pregnancy characteristics on fetal epigenetics and later development.

While analysis of miRNA is not a new subject, authors presented well designed study and robust analysis of the data. Authors are aware of the limitations of their data and need for further research to include more subjects. based on available numbers, well thought conclusions are presented.

Easy to read language and paper follows easy to understand story.

END OF COMMENTS

Reviewing Editor's comments:

Graphical abstract - Should the text at the bottom of the figure use the terms 'responsive to' rather than 'sensitive towards'?

Authors: The text at the bottom of the figure was now changed to: *Circulating umbilical cord plasma miRNAs profiles are shaped by maternal and neonatal metabolic traits.*

The introduction is very brief. The rationale of the role of miR in relation to maternal BMI could be expanded.

Authors: As suggested by Reviewer 1, we have added a paragraph explaining how fetal sex could influence materno-fetal communication as the third paragraph in the introduction. The second paragraph has been rearranged to more clearly explain the relationship between maternal BMI and UCP miRNAs. The two paragraphs read:

Maternal metabolism profoundly influences fetal metabolism and development by regulating the delivery of key nutrients, such as glucose and lipids. For instance, maternal overweight and obesity are associated with higher birth weight, increased neonatal body fat (Friis et al., 2013, Wan et al., 2023), and fetal insulin resistance (Catalano et al., 2009). Conversely, pregnancy presents a challenge to the maternal metabolism, initiating a series of metabolic adaptations driven by pregnancy hormones, growth factors, and other regulatory molecules, many of which are released by the placenta (Stern et al., 2021). These bioactive factors play a crucial role in adapting maternal metabolism to meet the demands of pregnancy and ensuring an adequate supply of nutrients to the developing fetus (Stern et al., 2021). As such, maternal BMI is likely one of the most significant factors influencing fetal development, fundamentally shaping feto-maternal communication, and potentially impacting UCP miRNA patterns.

Interestingly, the release of various placenta-derived hormones and growth factors differs depending on fetal sex, leading to distinct levels of certain hormones and growth factors, such as progesterone, placental growth hormone and human chorionic gonadotropin, in the maternal circulation (Stern et al., 2021). Moreover, some bioactive factors in the fetal circulation, such as leptin (Tome et al., 1997), also vary between the sexes. Most notably, neonatal fat mass significantly differs between male and female neonates (Rodriguez et al., 2004, Shields et al., 2006), positioning fetal sex as a potential factor influencing feto-maternal communication.

E: Line 89 - UCB was collected from healthy pregnancies. Were a range of maternal BMIs included? How was this determined as only 16 samples were collected? Was there an a priori decision to collect from an equal number of male and female pregnancies? How many placentas were collected at each BMI?

Authors: These questions are now being addressed in the methods section. It reads:

Venous cord blood was collected from 40 healthy full term pregnancies. Umbilical cord blood of 16 women was selected in order to achieve a wide range of maternal pre-pregnancy BMI, i.e., 8 women with normal pre-pregnancy BMI (<25), and 8 women with a high pre-pregnancy BMI (>25). Moreover, we selected women in order to include a similar number of pregnancies with male and female fetuses in both (normal and high pre-pregnancy BMI) groups. Cord blood was drawn immediately after delivery using Vacuette EDTA blood collection tubes (Greiner Bio-One, Kremsmünster, Austria). Plasma was obtained after centrifugation at 3500 rpm and 4°C for 10 min and stored at -80°C for analysis.

For visualization of the BMI distribution (pre-pregnancy and at birth) of our cohort, we have included violin plots of them below:

E: Was higher maternal age associated with higher maternal BMI? If yes, may this impact the findings?

Authors: There was no correlation between pre-pregnancy BMI and maternal age ($R=-0.103$; $p=0.704$) and BMI at birth and maternal age ($R=0.075$; $p=0.792$).

Referee #1:

Overall, the work by Strutz et al., reads well and provides insight into UCP miRNAs and their association with maternal and fetal morphometric measures as well as some markers of metabolic health.

Authors: We thank the reviewer for thoroughly reviewing and improving our manuscript. We hope to have addressed all comments appropriately.

Introduction

R: The introduction is very succinct; however, based on the layout of Table 1, it would make sense to expand on how fetoplacental responses are different between sexes in normal and overweight pregnancies.

Authors: As suggested by the reviewer we included a paragraph in the introduction on how fetal sex affects feto-maternal communication. It reads:

Interestingly, the release of various placenta-derived hormones and growth factors differs depending on fetal sex, leading to distinct levels of certain hormones and growth factors, such as progesterone, placental growth hormone and human chorionic gonadotropin, in the maternal circulation (Stern et al., 2021). Moreover, some bioactive factors in the fetal circulation, such as leptin (Tome et al., 1997), also vary between the sexes. Most notably, neonatal fat mass significantly differs between male and female neonates (Rodriguez et al., 2004, Shields et al., 2006), positioning fetal sex as a potential factor influencing feto-maternal communication.

Methods

R: Were any of the maternal or neonatal factors included as covariates in the miRNA analysis pipeline?

Authors: Due to our small sample size, we did not include any factors as covariates in the analysis. Instead, we focused on balancing the cohort as much as possible from the outset (e.g., 4 male and 4 female newborns in the overweight pregnancies; 4 male and 4 female in the normal-weight pregnancies). However, to evaluate whether neonatal sex might influence UCP miRNA profiles, we identified miRNAs exhibiting sex dimorphism and analyzed their overlap with correlations to maternal or fetal characteristics (Table 6). There was no overlap with miRNAs correlating to maternal pre-pregnancy BMI or BMI at birth.

Results

R: Table 1. Can the authors comment on the rationale for displaying the data split by sex, if the focus of the study was to analyse UCP miRNAs in relation to maternal BMI? It would

make more sense to compare outcomes by BMI throughout the study and, where appropriate, stratify by sex.

Authors: We agree with the reviewer. We gave too much emphasis to sex differences because it is a topic that generally interests us. However, in this study, it was only a side aspect. The sex differences in the subject characteristics are now included in Table 5, but are no longer part of the general subject characteristics table (Table 1). Since the subgroup sizes in our cohort are quite small due to the sex distribution, we did not stratify by fetal sex. As already mentioned above, we were aware of this fact and balanced the cohort as much as possible from the outset.

R: UCP miRNA profiling. The authors state that they focussed on miRNAs with a mean CPM >100 - what was the rationale for using this cut off? Considering many (~90%) of miRNAs had a CPM <100, is it likely that biologically meaningful findings have inadvertently been overlooked based on these parameters?

Authors: It is true that miRNAs with lower counts can also have a biological effect. Therefore, we repeated the entire data analysis, now including all miRNAs above the threshold of 10 CPM, and modified the entire manuscript accordingly. While number of correlating miRNAs increased, and also the proportion of miRNAs attributed to the individual characteristics changed slightly, but most importantly, the main findings of the study remained consistent with the prior analysis using a threshold of 100 CPM.

R: 'Distinct UCP miRNA signatures in male vs. female newborns' - are these signatures unique to normal or overweight BMI pregnancies? Have similar analyses been performed between male <25 vs >25 BMI and female <25 vs >25 BMI pregnancies? This would certainly be interesting to examine.

Authors: 'Distinct UCP miRNA signatures in male vs. female newborns' refers to all male vs all female neonates, regardless of maternal BMI. We agree with the reviewer is that it would be interesting to see if the sex differences occur in normal weight pregnancies or in overweight pregnancies, or both. Unfortunately, the group size (n=4 male and n=4 female UCP) in these subgroups is not large enough to perform these analyses and draw reliable conclusions.

Discussion

R: The **sex-specific differences section** of the discussion could expand to include any known role of **these miRNAs in the regulation of genes required for placental development/fetal growth**.

Authors: In this revised version, we performed a pathway analysis on the miRNAs that exhibited sex differences. The following text was added to the Results and Discussion sections:

Results: *Pathway analysis of miRNAs with sex differences using miRPath 4.0 online tool based on TarBase and TargetScan revealed 'cell cycle', 'p53 signaling pathway' and 'ubiquitin mediated proteolysis' as the top three pathways.*

Discussion: *Pathway analysis suggested that miRNAs with sex differences may be related to growth and tissue turnover. Infact, studies based on larger cohorts revealed higher birth weight of male neonates (Voskamp et al., 2020) which may be reflected by the UCP miRNA profile.*

Referee #2:

Thank you for submitting your paper.

The results presented in the article is opening new area of research and contributes to understanding of the physiology and pathology of interaction between fetus, placenta and mother. The analysis of miRNA cast light on molecular interactions and impact of pregnancy characteristics on fetal epigenetics and later development.

While analysis of miRNA is not a new subject, authors presented well designed study and robust analysis of the data. Authors are aware of the limitations of their data and need for further research to include more subjects. based on available numbers, well thought conclusions are presented.

Easy to read language and paper follows easy to understand story.

Authors: We thank the reviewer for the positive feedback.

Dear Associate Professor Hiden,

Re: JP-RP-2025-287672R1 "MiRNA Profiling in Umbilical Cord Plasma: Links to Maternal Metabolism and Neonatal Metabolic and Inflammatory Traits" by Jasmin Strutz, Anna Rieder, Carolina Tocantins, Mariana S. Diniz, Elisa Weiss, and Ursula Hiden

We are pleased to tell you that your paper has been accepted for publication in The Journal of Physiology.

Yours sincerely,

Laura Bennet
Senior Editor
The Journal of Physiology

If you would like to receive our 'Research Roundup', a monthly newsletter highlighting the cutting-edge research published in The Physiological Society's family of journals (The Journal of Physiology, Experimental Physiology, Physiological Reports, The Journal of Nutritional Physiology and The Journal of Precision Medicine: Health and Disease), please click this link, fill in your name and email address and select 'Research Roundup':
<https://www.physoc.org/journals-and-media/membernews>

- You can help your research get the attention it deserves! Check out Wiley's free Promotion Guide for best-practice recommendations for promoting your work at: www.wileyauthors.com/eoo/guide. You can learn more about Wiley Editing Services which offers professional video, design, and writing services to create shareable video abstracts, infographics, conference posters, lay summaries, and research news stories for your research at: www.wileyauthors.com/eoo/promotion.

Reviewing Editor's comments:

Thank you for acknowledging the reviewers' feedback and revising the paper.

Referee #1:

The authors have addressed all previous comments.

Referee #2:

Thank you very much for sending reviewed article. Changes improved readability and made the paper easy to understand.

END OF COMMENTS